# Social Inclusion of Transgender People in Intercollegiate Sports—A Scoping Review

Liliana Mendes [1,*], Elsa Gabriel Morgado [2] and Levi Leonido [3,4]

1   Coimbra Institute for Biomedical Imaging and Translational Research, University of Coimbra, 3000-548 Coimbra, Portugal
2   Center in Basic Education, Polytechnic Institute of Bragança, 5300-253 Bragança, Portugal; elsa.morgado@ipb.pt
3   Portuguese Catholic University-CITAR, 4169-005 Porto, Portugal; levileon@utad.pt
4   University of Trás-os-Montes e Alto Douro, 5000-801 Vila Real, Portugal
*   Correspondence: liliana.mendes@icnas.uc.pt

**Abstract:** Transgender individuals face discrimination and exclusion in various areas of society, including sports. Notably, intercollegiate athletics suffer criticism for their lack of inclusivity towards transgender athletes. Despite the increasing visibility of transgender individuals and ongoing efforts towards greater inclusivity, there is a significant lack of research on their social integration within college sports. This scoping review aimed to explore the then-current state of research on the social integration of transgender individuals in intercollegiate athletics, identify gaps in the literature, and suggest areas for further investigation. The study examined articles published between 2013 and 2023, using databases such as PubMed, ERIC, and EBSCO Essentials, as well as relevant citations from selected articles. The inclusion criteria for articles were their focus on the social integration of transgender individuals in sports events, publication in English, and relevance to the research question. A standardized technique based on the PRISMA flow diagram 2020 was used to locate, assess, and extract information from eligible research. The results of the study are expected to inform policy and strategy in transgender participation in college sports and promote greater inclusivity for transgender individuals in sports institutions and groups.

**Keywords:** cultural identity; education; gender equality; social inclusion; social inequalities; transgender

## 1. Introduction

According to the World Health Organization (WHO), transgender individuals have an incompatibility between the gender given to them at birth with their gender identity. The gender identity of a person is closely associated with the social transition. A social transition is a means of altering one's physical attributes, behavior, as well as societal position to more closely correspond with one's gender identification without the need for medical procedures (WHO 2022). This might include wearing gender-specific clothes and haircuts, altering their name, style, and behaviors, and utilizing gender-specific services such as toilets and restrooms, which match their gender identification. Many transgender persons will opt for social transition to validate their gender identification, while others will choose to physically transition using gender-affirming hormones and gender-affirming surgery.

For most nations, transgender persons are excluded and marginalized due to a dearth of state recognition (Weiselberg et al. 2019). Though several nations now accept a "third gender," transgender persons are sometimes compelled to endure genital surgery to acquire the official status of the gender; also, some nations would never acknowledge a change in a person's birth gender (WHO 2022). The WHO report additionally states that, overall, transgender persons have poor levels of access to medical treatments because of a variety of

factors such as violence against them, legal hurdles, stigma, and prejudice. Transgender persons are frequently abused (WHO 2022). Moreover, transgender persons may face rejection from their families as well as violations of their privileges to education, work, and social benefits (Mountjoy et al. 2016). As a result, transgender individuals may face increased levels of unemployment, suffering, homelessness, and marginalization (WHO 2022). This situation is similar in the sports environment.

Sport constitutes a highly gendered environment in which conventional concepts of masculinity, as well as femininity, are frequently reinforced and maintained. In comparison to the sports played by males and females, sports for transgender people have been under-rated, under-promoted, and transgender athletes are frequently exposed to objectification and criticism about their physical traits. Furthermore, there has long been controversy regarding whether trans athletes deserve to be permitted to participate within gendered categories, which correspond to their gender identification (Mountjoy et al. 2016). The binary concepts of sex and gender, which are often utilized in athletics, provide difficulties for trans persons who are unable to fit properly into these classifications. For example, according to a recent investigation, many believe transgender women retain an athletic edge over their cisgender counterparts despite spending a year on gender-affirming therapy (Roberts et al. 2020). Therefore, sometimes transgender women athletes might not be permitted to participate in a collegiate sports event, or they might have to face additional difficulties (Roberts et al. 2020). However, the Canadian Center for Ethics in Sport (CCES 2022) claimed that there has been insufficient evidence to support the influence of testosterone suppression on the athletic abilities of transgender women athletes. According to the research, trans women who have had testosterone suppressed have no evident biological benefits over cis women in competitive athletics (CCES 2022). Additionally, discriminatory rules and views against trans athletes may pose hurdles to their engagement within collegiate sports and undermine their feeling of being accepted in these areas. Trans athletes could feel alone (DeFoor et al. 2018), misinterpreted, and ignored by their teammates and coaches, leading to reduced involvement as well as a rise in psychological difficulties. However, it is critical to acknowledge that gender can be a complicated and varied concept, as well as that trans athletes ought to be honored with decency, respect, and equality in a collegiate setting. This necessitates a change from the binary gender paradigms into broader and more flexible regulations that accommodate a variety of gender orientations and manifestations within collegiate sports (DeFoor et al. 2018). Regulations such as these must be founded on equality and inclusion rules, instead of on strict and old-fashioned gender stereotypes that restrict and differentiate against trans athletes (DeFoor et al. 2018).

Additionally, transgender social inclusion throughout sports, especially intercollegiate athletics, represents an important and topical subject that has received a greater spotlight throughout recent years. Despite considerable advances, transgender people continue to encounter major impediments related to full involvement in athletics because of their gender identification (Kamasz 2018). These restrictions are not confined to collegiate athletics; in some nations, transgender people have been barred from participating in sports or compelled to participate in activities that do not correspond to their gender identification (Stewart et al. 2018). Nevertheless, some of the particular obstacles encountered by transgender athletes within the context of intercollegiate settings might involve the insufficient availability of gender-affirming medical care as well as amenities, in addition to a dearth of sympathy and encouragement from coaches, colleagues, as well as athletic groups. These exclusions may have substantial psychological and physical effects on transgender people, as well as raise instances of prejudice against them (Westbrook and Schilt 2014). Overcoming these issues is not simply a problem of human rights; it is also an issue of public health. Therefore, it is critical that measures be taken to foster social inclusion as well as encouragement for transgender athletes throughout collegiate settings (Westbrook and Schilt 2014).

An international effort also has been made to reduce the bias against transgender people's sports participation. In 2004, the International Olympic Committee (IOC) announced

that transgender individuals who have undergone total medical conversion may participate in all subsequent Olympic Games (Devine 2022). This decision aimed to make it easier for transgender athletes to compete at the highest level of sport. The IOC has amended its rule to become more accepting of transgender athletes; also, the 2004 guideline has had a significant influence on the policy-making of other sports organizations (Genel 2017). Trans women could benefit from a competitive advantage within sports after the medical transition because of earlier exposure to testosterone hormones (Devine 2022), which might confer some physical benefits such as enhanced bone density, muscle mass, and increased oxygen-carrying ability. Even though the policy's standards seem to back the widely held idea that transgender persons have an athletic edge, they have already been criticized for lacking an evidence-based basis. However, the IOC currently does not have a two-year minimum. In establishing these guidelines, the IOC believes it is the responsibility of each sport including its governing organization to decide how a participant may have a disproportionate advantage over other athletes, considering the characteristics of every sport (Mountjoy et al. 2016). Nonetheless, there seems to be a significant gap in the rights and restrictions enforced on trans athletes who consider themselves male vs. those who consider themselves female. It implies that the consideration of trans athletes is not constantly uniform or equal throughout gender identities (Kondareva 2019). Altogether, this may give a rough idea of the hardship faced by transgender athletes. Discussing the IOC's stance on this topic is particularly important because trans athletes who want to participate in collegiate sports may also want to take part in international sports events in the future. Therefore, it might act as a negative motivator for a transgender college student who thinks about taking part in college sports (Kondareva 2019). Nevertheless, the National Collegiate Athletic Association (NCAA) has offered support for transgender student athletes, numerous athletes continue to suffer difficulties and prejudice. Regardless of their promises of inclusivity, many organizations and sporting events maintain discriminatory rules and views regarding trans athletes (Kondareva 2019). The recent restriction on trans women competing at the highest level issued by World Athletics/Rugby Football Union (RFU) adds to the prejudice encountered by trans athletes across sports. According to BBC News, the Rugby Football League and RFU have also prohibited transgender people from participating in female-only versions of their sports. It came after World Rugby became the first international sports federation to state in 2020 that transgender females cannot play at the highest and international levels of female sports (Falkingham 2023). However, the NCAA guideline is more comprehensive than the IOC's completed rules (Kondareva 2019). The NCAA does not necessitate sex reassignment surgery nor official acknowledgment of one's gender identification since it believes that a gender-affirming procedure is sufficient to neutralize as well as erase the gender-based physiological sports edge.

Moreover, meeting the needs of transgender individuals in collegiate sports is one aspect of diversity, which has received inadequate funding and attention. Researchers who examined inclusiveness and diversity initiatives involved in collegiate sports discovered few efforts that addressed the requirements of this group (DeFoor et al. 2018; Jeanes et al. 2019). Shaw (2019) calls sport's resistance to wider societal transformations linked to trans people an "enigma," whereas Cunningham and Hussain (2020) call it a "paradox" since on the one side, prejudice and bigotry hinder accessibility and possibilities for trans sportsmen.

Despite the rising acceptance of gender diversity as well as transgender rights, the sports industry remains sluggish in enacting inclusive guidelines and procedures, particularly regarding legal frameworks as on-the-ground execution. A similar condition can also be observed in collegiate sports settings. As a result, there seems to be an urgent need to investigate how to push activities towards more diversity and inclusion within collegiate sports (Denison et al. 2021). According to Robertson et al. (2019), trans diversification has been "missing" among recent talks throughout the sports leadership literature about challenges and opposition to diversification. Ellis et al. (2014) discovered that transgender people frequently avoid exercise and sports settings such as the gym in collegiate setting

because they are afraid of being harassed, classified as trans, or exposed to a negative categorization. Because gyms are a popular location for sport-related physical activity, it is critical that they promote an inclusive environment. Sports and exercise have been shown to have various mental and physical benefits, emphasizing the need for making people of all genders and orientations feel included and encouraged in these settings (Ellis et al. 2014). Firstly, transgender athletes' experiences in college athletics could provide light on the greater topic of transgender rights and equality. Secondly, studying the perspectives of transgender people in college athletics may help to shape guidelines and procedures, which encourage inclusiveness and enhance transgender athletes' health and well-being. Lastly, research on this issue may help to design evidence-based approaches to promote social inclusion as well as reduce stigma and prejudice towards the transgender population. Therefore, this scoping review analyses the current literature to understand the present situation of transgender people's social inclusion in intercollegiate sports. In addition, it will shed some light on findings gaps from the previous literature where further research is needed. Therefore, this research aims to investigate the current status of research on the social integration of transgender persons in college athletics.

## 2. Method

### 2.1. Study Design

A scoping review seems to be a study approach that seeks to outline the available literature on a certain issue to provide a complete overview of the present state of understanding (Munn et al. 2018). Scoping reviews are frequently used to spot gaps in the existing literature and to help shape the structure of future research investigations. However, it is different than the systematic review (Munn et al. 2018). Scoping reviews are commonly employed in the early phases of a research study to determine the volume, scope, and character of accessible literature on a particular subject. They cover a larger range of topics and are less comprehensive compared to systematic reviews. However, it can be carried out quickly and provides a border understanding of the topic (Munn et al. 2018).

A scoping review would be an appropriate method for studying the social inclusion of transgender individuals in intercollegiate athletics, given the need for a comprehensive understanding of the current literature on the subject, including the types of research conducted, populations studied, and outcomes measured. Furthermore, only a limited number of research articles focus on intercollegiate sports. Therefore, a scoping review can be considered a crucial first step in exploring this broad topic.

### 2.2. Search Strategy

A comprehensive literature search was conducted on various databases to identify relevant articles. To conduct this search, the key terms "Social Inclusion", "Social Integration", "Transgender", "Trans", "Intercollegiate Sports", and "College Game" were used. To develop a thorough search strategy, the terms would be merged using the Boolean operators AND, and OR. For example: (Social Inclusion OR Social Integration) AND (Transgender OR Trans) AND (Intercollegiate Sports OR College Game).

### 2.3. Database Description

A comprehensive literature search was conducted on PubMed, ERIC, and EBSCO Essentials. To find relevant papers, the above-mentioned keywords were used. In addition, the references of the selected articles were considered to extract more data.

### 2.4. Inclusion and Exclusion Criteria

Strict inclusion and exclusion criteria were developed to limit the number of relevant articles. For this research, only peer-reviewed papers were considered. The papers which were published from 2013–2023, written in English, accessible, and relevant to the research aim which is to investigate the current status of research on the social integration of transgender persons in college athletics, were included in this review. Papers that are older

than 2013, not relevant to the study question, and not written in English were excluded from this review. This study focuses on analyzing more current and recent papers to draw up-to-date findings, hence, a paper published before 2013 were not chosen.

### 2.5. PRISMA Design

PRISMA is an acronym that refers to Preferred Reporting Items for Systematic Reviews and Meta-Analyses. It is the bare essential set of evidence-based elements for inclusion during systematic reviews or meta-analyses (Page et al. 2021). A PRISMA flow diagram was used to identify relevant papers. The screening procedure is summarized in the PRISMA flow chart (Figure 1). It first counts the number of papers identified before making the selection procedure visible by reflecting on choices made at different phases of the scoping review. At each level, the number of publications is noted (Page et al. 2021). For this study, this framework can be helpful to identify the most relevant articles, as per the research question.

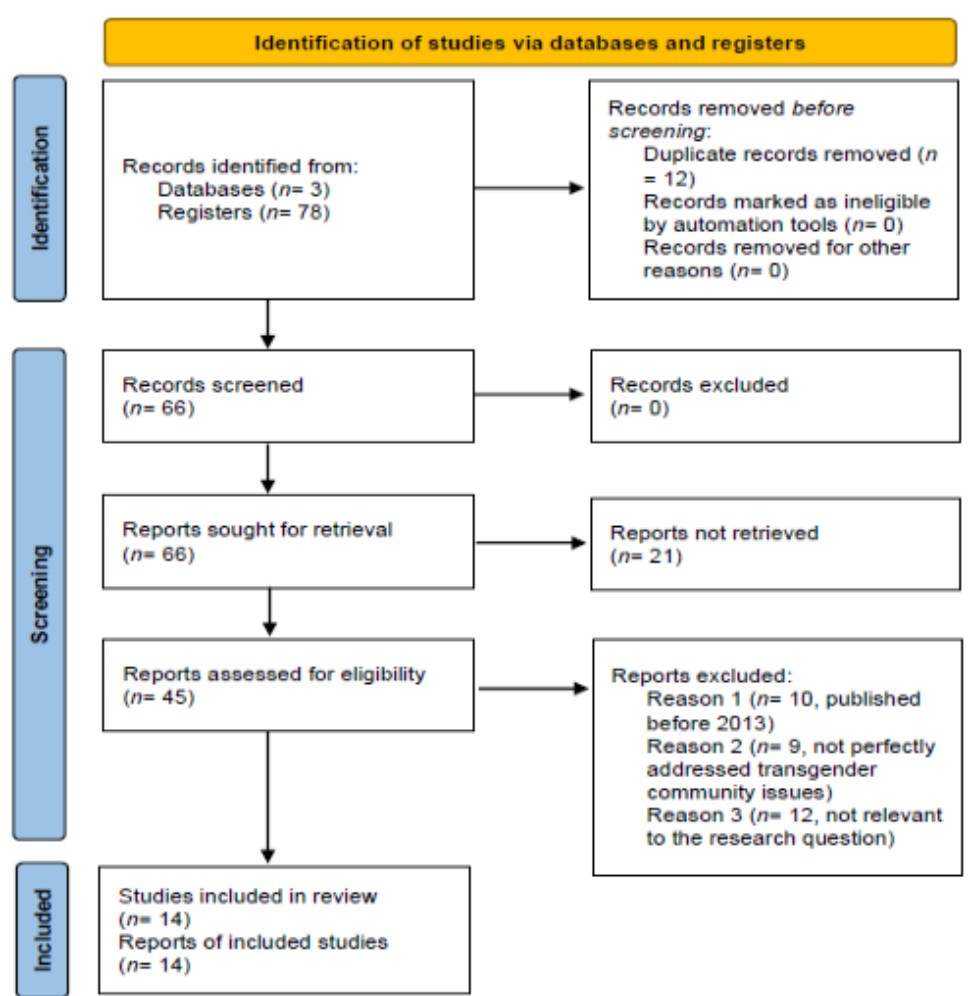

**Figure 1.** PRISMA flow chart (Page et al. 2021).

### 2.6. Other Tools

Other than the PRISMA flow chart, Microsoft Excel was used. Microsoft Excel has calculation and computation abilities, graphing functions, as well as abilities to create pivot tables. It is a valid and reliable tool to generate graphs (Divisi et al. 2017). Here, Microsoft Excel was used to generate the graphs.

### 3. Results and Data Extraction

*3.1. Selection Based on the PRISMA Flow Chart*

Based on the initial search of three databases (PubMed, ERIC, and EBSCO Essentials) a total of 78 papers were identified, among them, 14 papers were finally selected (Figure 1). During the initial screening stage, after adjusting the duplication studies, a total of 12 studies were removed. Additionally, 21 additional papers were removed since those papers were not retrievable. This led to 45 papers remaining. Among these 45 papers, inclusion and exclusion criteria were applied to carry out the final screening. Ten papers were removed since they are published before 2013; nine papers were removed since the topic is not specifically related to the transgender or trans community; and 12 papers were also removed since they are not specific to the research question, which was the inclusion of transgender people in sports. This led to a final selection of 14 papers, based on which the analysis was made. The data analysis was carried out by manually downloading and reading the full paper to gather relevant information.

*3.2. Publication Year*

Among the selected papers, most of them (three papers) were published in 2018. Additionally, a total of five papers were published within the timeframe 2013–2017, and six papers were published between 2019–2023 (Figure 2). Hence, it can be stated that the papers used in this analysis were relevant and current.

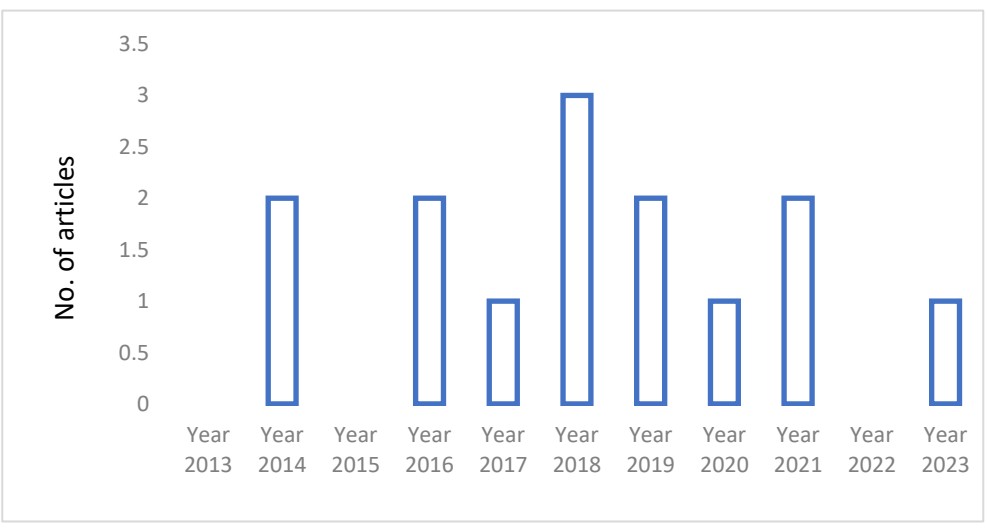

**Figure 2.** Year-wise analysis.

*3.3. Study Type*

As indicated in Figure 3, much of the study (*n* = 9) comes from the primary studies. In addition, this review paper contains five literature review papers, which provide primary evidence for the scoping review.

The literature reviews include review papers, systematic reviews, and normal literature analyses. All these papers provide a great in-depth understanding of the current stage of social inclusion of transgender people in sports.

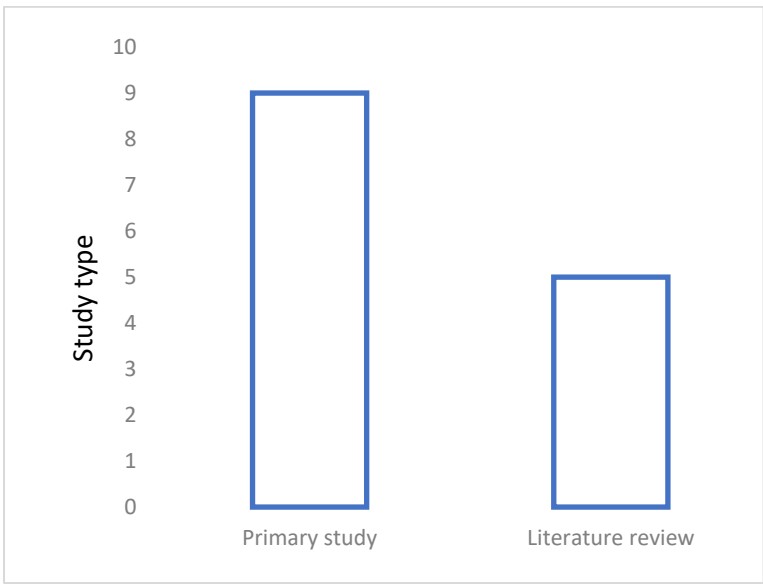

**Figure 3.** Type-wise analysis.

*3.4. Literature Table*

This section summarizes the main findings of the selected articles to provide context and clarity surrounding relevant issues related to the relationship between social inclusions of transgender students in intercollegiate sports. Relevant articles were included in this review (Table 1).

**Table 1.** Summary of the key findings of the selected articles.

| Citation | Main Finding |
|---|---|
| Ellis et al. (2014) | Transgender persons avoid certain circumstances to avoid being revealed (primary study). |
| Waselewski et al. (2023) | Many people believed that inclusive laws could reinforce and promote transgender people's mental wellness. A few participants reported negative effects on fairness (primary study). |
| Jones et al. (2017) | Transgender persons have a largely poor experience within competitive sports due to the limits imposed by the sport's regulation (systematic literature review study). |
| Huffaker and Kwon (2016) | Sexual discrimination research includes views toward various subgroups inside the greater LGB and transgender population (literature review study). |
| Cunningham and Pickett (2018) | According to the conclusions of this study, while prejudice towards trans people has lessened, more programs and bias reduction initiatives are required (primary study). |
| Tanimoto and Miwa (2021) | Transgender athletes are more welcomed in leisure sports as opposed to in competitive tournaments (primary study). |
| Cunningham et al. (2018) | The authors make the argument for an inclusive locker or dressing rooms, which allow transgender individuals access to amenities, which match their gender identification (literature review study). |
| Reynolds and Hamidian Jahromi (2021) | The author acknowledges it is an underexplored topic with no straight solution (literature review study). |
| Bassett et al. (2020) | The International Olympic Committee (IOC) presently requires female athletes having sex development variations, or intersex features, as well as a transgender female sportsman to restrict their plasma testosterone below <10 nmol/L for one year in order to compete inside the female category (literature review study). |
| Phipps (2021) | Feedback from trans students, along with officer perspectives, show that more effort may be made to guarantee that university athletics is accessible to all, especially with reference to the dependence on larger binary gender systems within sport (primary study). |

**Table 1.** *Cont.*

| Citation | Main Finding |
| --- | --- |
| Klein et al. (2019) | The influence of strict compliance to the binary sex as well as gender within sport, in addition to the way ordinary gendered activities create hurdles for trans athletes, was underlined through the conceptual framework (primary study). |
| Stroup et al. (2014) | Due to their surroundings, gender minority students in remote universities throughout the United States might face significant challenges (primary study). |
| Atteberry-Ash et al. (2018) | Campus athletic facilities could be inhospitable to trans students (primary study). |
| Toomey et al. (2016) | The participation throughout trans students' ally activities within the sports scale revealed good internal consistency, accuracy, as well as equivalence throughout gender (primary study). |

*3.5. Key Findings*

- Prejudice and discrimination remain a key issue for intercollegiate sports participation.
- Policies governing college sports participation are uneven, creating a barrier for trans athletes.
- Transgender students have faced a lack of support and shared space challenges, which prevent them from participating in intercollegiate athletics.
- The heteronormative concept, as well as the binary structure of sports, act as a barrier to intercollegiate sports participation.

**4. Discussion: Description and Analysis**

*4.1. Prejudice, and Discrimination in College Settings*

According to a systematic review, Jones et al. (2017) state that transgender athletes within collegiate sports encounter prejudice and bias due to their gender identification, which may have a detrimental influence on their involvement in sports including athletic performance. This prejudice manifests itself in a variety of ways, such as rejection from teams, bullying, and discrimination. This may lead to transgender persons avoiding specific circumstances related to sports. Ellis et al. (2014) investigated contextual avoidance with special regard to gender identification as well as the phase of transition using a population of 889 UK-based individuals who self-identified as transgender. This study discovered statistically significant connections among subgroups (gender identification as well as phase of transition) as well as the avoidance (or otherwise) of specific circumstances, such as clothing stores, collage gyms, and restrooms. The significance of these results for assisting transgender persons in transition, particularly the actual experience, is indeed highlighted. This finding suggests that trans athletes may face difficulties in collegiate settings (Ellis et al. 2014; Jones et al. 2017).

Similar to Ellis et al. (2014), Klein et al. (2019) also mention a significant level of stereotyping against transgender people participating in college sports. According to Klein et al. (2019), many trans athletes have suffered dissension and hatred from teammates and strangers. Some peers, opponents, or officials, for example, routinely used wrong pronouns or openly attacked trans athletes. Many trans athletes were forbidden from competing. Additionally, Jones et al. (2017) acknowledged the concerns and obstacles that transgender individuals face when engaged in athletic competition as well as sport-related exercise. This discrimination and exploitation is the result of preconceptions articulated via power dynamics. To avoid bias and discrimination, trans student athletes are frequently hesitant to reveal their sexual identity, and therefore feel obligated to employ orientation negotiation or impression control strategies (Jones et al. 2017).

Additionally, according to Phipps (2021), using existing academic research, considerable impediments remain in collegiate sport, a setting that is largely constructed with dependence on gender binaries. Because of the gender binary, collegiate sports venues tend to provide substantial challenges for trans athletes. Concerns about eligibility remain a significant barrier, as many organizations rely on outmoded procedures, which do not

take gender identity into consideration. Transphobia is another major issue, with numerous athletes experiencing prejudice and abuse from colleagues, instructors, and followers (Cunningham and Pickett 2018). The gender binary fosters the notion that there are just two genders, which are concrete and permanent. This fosters a hostile atmosphere for trans competitors, who are frequently denied equitable access to games and suffer participation difficulties. As a consequence, trans athletes face significant challenges to participate within collegiate sports situations (Cunningham et al. 2018; Jones et al. 2017).

Another form of prejudice encountered by transgender athletes within collegiate sports includes stigmatization. This stigma stems from societal prejudices and misconceptions regarding transgender people, such as the assumption that they possess an unfair edge in athletics (Reynolds and Hamidian Jahromi 2021). This judgment may result in transgender athletes being excluded from athletic groups and tournaments, even if they are capable of competing at a college level. Discrimination against transgender athletes within collegiate or university sports may result in serious consequences (Waselewski et al. 2023). Discrimination may trigger low self-worth, nervousness, sadness, and other psychological conditions that might impair sports performance. Discrimination may additionally lead to a dearth of opportunity and assets, limiting their potential to excel in athletics (Huffaker and Kwon 2016). Additionally, transgender studies have revealed that opinions about trans people are mostly negative among cisgender people (Huffaker and Kwon 2016). Even though with recent exposure and policy changes, Cunningham and Pickett (2018) mention the condition is currently slightly improving, but still the rate of prejudice and racism against trans people is significantly high, mainly considering participation in sports. Similarly, trans individuals encounter considerable obstacles linked to bias and prejudice within collegiate sports situations. These difficulties are caused by several issues, notably societal stigma as well as an absence of knowledge, along with legislation and procedures, which are frequently discriminatory against trans athletes (Reynolds and Hamidian Jahromi 2021; Tanimoto and Miwa 2021). One of the major concerns is the absence of straightforward and uniform regulations addressing trans athletes' involvement in collegiate sports. It may contribute to inconsistencies and uncertainty regarding the way trans athletes are perceived, as well as hurdles to their engagement in sports (Cunningham et al. 2018; Stroup et al. 2014). Some colleges, for instance, might ask trans athletes to undergo gender-affirming procedures for a set amount of time before they may participate, whilst others might not. This inconsistency may make sports participation tough and unjust for trans athletes (Atteberry-Ash et al. 2018).

Additionally, prejudices can be specific to a particular transgender community. Tanimoto and Miwa (2021) mention prejudices are more prevalent against the transgender women community in sports. The author collected quantitative data from 373 Japanese students. The findings indicate that transgender men were far more recognized than transgender women; transgender athletes receiving gender-affirming hormonal therapy were more allowable, and such transgender sportsmen were more acknowledged in non-official national and global sports competitions for adolescents and adults compared to formal national as well as international activities. Tanimoto and Miwa (2021) also mention higher levels of confidence in a fair society were also linked positively with accepting views among participants having weaker sports personalities. Greater athletic personality was favorably connected with male acceptability but adversely linked with female acceptability.

Summary: This theme emphasizes the presence of prejudice and discrimination against transgender people in sports participation. The examination of other areas of literature (including non-college competitive sports), as well as literature regarding collegiate settings, reveals a high probability of racism towards trans individuals in collegiate settings, which might operate as a barrier to their participation in intercollegiate sports.

### 4.2. Policy Barrires for Intercollegiate Sports Participation

Several regulations and measures have been created to tackle the prejudice and problems experienced by transgender athletes within college sports. The National Collegiate

Athletic Association's (NCAA) policy on transgender athlete engagement is one of these rules. This regulation permits transgender athletes to compete in activities depending on their gender identification instead of their biological sex. The rule intends to provide transgender athletes with equal chances and to safeguard them from prejudice (Phipps 2021). However, others have contended that this measure is insufficient for safeguarding transgender athletes' rights and prospects. One of the most common critiques leveled at the NCAA regulation is that it places too much emphasis on one-year gender-affirming treatment as a requirement for eligibility (Reynolds and Hamidian Jahromi 2021). In this case, it is important to consider Lia Thomas' case. Thomas' transfer from the male group to the female squad is the consequence of her gender transformation, and after completing the NCAA's one-year testosterone suppressant usage criteria, she is able to play in collegiate sport as a part of a female team. According to an editorial in the University of Detroit Jesuit, Szachta (2022) mentions that the University of Pennsylvania is plainly breaching Title IX by enabling Lia Thomas to compete in female swimming and depriving biological women of opportunity. The existing eligibility standards for transgender athletes jeopardize the possibilities of female athletes because the NCAA simply mandates that the athletes' testosterone levels do not go above 10 nmol/L. Biological males possess a significant physical benefit compared to females, and no level of hormone treatment will eliminate physical benefits such as height and muscle development, as seen by Lia Thomas' exceptionally strong and tall body (Szachta 2022)

The present controversy about allowing transgender athletes to compete in athletics (in their present form) is concentrated on biological distinctions, most particularly, those between transgender as well as cisgender women. Skill differences depending on "assigned sexual orientation at birth" differ throughout sports, with swimming having the least and running and track activities having the greatest (Bassett et al. 2020). Such variations throughout athletic performance do not manifest themselves until after maturity and are assumed to be caused by higher circulating levels of testosterone among the "male" identified gender at birth players relative to the "female" assigned gender at birth players (Reynolds and Hamidian Jahromi 2021). Nevertheless, there has been a general paucity of evidence linking increased levels of testosterone to enhanced sports performance. The physical benefits, which transgender athletes may have, are at the heart of the controversy about their participation in sporting events. Additionally, the "female" categorization in athletics is quite unclear and just not universally recognized. To keep events fair, standardized norms regulating the admission or expulsion of transgender athletes must be developed. Presently, policies and perceptions of participation vary greatly at the collegiate sports level of activity (Tanimoto and Miwa 2021), implying that transgender players are seen differently at the amateur as well as professional levels (Cunningham et al. 2018; Tanimoto and Miwa 2021). Hence, both the scientific and medical societies must contribute to the development of such standards, particularly related to gender-affirming approaches and more so, the relationship between levels of testosterone and enhanced sports performance. Although physicians should play a significant role in formulating new athletics policies, sporting events managers mainly in collegiate settings and individuals who have expertise in sports administration and development must also be recognized. Opening dialogue amongst all these people is the initial step toward assuring the implementation and adoption of new rules at collegiate athletic competition (Reynolds and Hamidian Jahromi 2021).

Summary: The theme focuses on the policy problem of transgender athlete participation within intercollegiate sports, which is a possible impediment owing to a lack of established rules governing their integration or dismissal. The NCAA's regulation allows transgender athletes to participate based on gender identity, but the one-year gender-affirming procedure mandate is insufficient to ensure equal opportunity, possibly providing transgender competitors an unfair edge. The college sports setting is distinct in that regulations regulating transgender athletes' involvement may vary greatly between schools,

resulting in erratic and unpredictability encounters for transgender athletes. As a result, transgender athletes may have difficulties participating in college sport competitions.

### 4.3. Shared Place Problem and Lack of Support in College Settings

Transgender athletes might confront an additional practical impediment in the absence of a secure and welcoming changing area (Cunningham et al. 2018). According to research, social issues within the collegiate sport setting might result in a reduced level of sporting involvement among transgender athletes (Cunningham et al. 2018). The very first issue is the locker rooms because many individuals might not feel at ease with their pre-surgery physique, fearing criticism from their contemporaries and being concerned about inconsistencies in the breast or genital area compared to everyone else (Waselewski et al. 2023).

Transgender women have stated the biggest obstacle to their involvement is a lack of an accommodating and comfortable atmosphere in the collegiate setting (Jones et al. 2017), which may explain their lower engagement in team sports. Transgender women regard their opinions, in particular, as a contributing impediment to their absence of engagement (Reynolds and Hamidian Jahromi 2021). Moreover, highly gendered sports create an atmosphere for these individuals that renders them fearful that opening up or applauding others might end up in them being incorrectly recognized as women (Reynolds and Hamidian Jahromi 2021). Likewise, considering sports clothing might be physically disclosing, it may pose an obstacle to engagement (Cunningham and Pickett 2018).

The shared space dilemma stems from the conflict between the requirement of offering athletes with an equal opportunity and the desire to encourage inclusion and diversity within athletics. On the one hand, athletic groups have to guarantee that all participants compete in a fair and equitable environment (Waselewski et al. 2023). The same is also true in the collegiate sports settings. This implies that athletes have to compete against those of the same sex allocated at birth, since physical disparities such as muscle mass as well as bone density might give an unfair advantage. This approach, nevertheless, could be discriminatory toward transgender athletes, who might not accept the sex given to them, indicating a lack of support to the transgender athletes (Cunningham et al. 2018).

Summary: The theme emphasizes that the shared space conundrum stems from a contradiction between giving equal opportunities for athletes and encouraging inclusiveness and diversity within athletics, which might be discriminatory against transgender athletes who do not identify with their assigned sex. Lack of encouragement and support from coaches, authorities, and other teammates may lead to transgender athletes dropping out of sports. It is possible that these transgender athletes are hesitant to engage in college sports due to this lack of support.

### 4.4. Other Stressors and Heteronormative Environment in College Settings

Stressors could be amplified for the first-year student athletes as they begin their college transition with the concern of injuries, inability to secure playtime, and failure to retain top athletic positions while shifting to a high level of competition (Phipps 2021). Furthermore, first-year student athletes encounter unique obstacles such as managing time and skipping class for squad travel, leading to less time or capacity to adjust to university life. It is straightforward to see that the pressures of competing for athletic ability at a top standard are amplified from the perspective of the trans student athlete, who has the added pressure to succeed and comply in a primarily heteronormative environment of athletics (Cunningham and Pickett 2018). Support networks and emotions of acceptability are increased areas of risk for transgender student athletes while shifting to a collegiate setting, as they also confront the difficulty of dealing with gender identity discrimination (Stroup et al. 2014).

Moreover, according to a study on trans inclusion sentiments among heterosexual students who participate in the club along with intercollegiate sports, a majority of students were neutral regarding the topic, with 35.5% expressing that they do not agree nor disagree that protective rules for trans athletes need to be set up; nearly 20% of student athletes

disagreed with the idea that such regulations need to be put within the spotlight (Atteberry-Ash et al. 2018). Female transgender, liberal political philosophy, and not meeting a trans sportsman are all characteristics related to greater heterosexual student athlete assistance for the trans-inclusive policies (Waselewski et al. 2023). Increasing perceptions of team tolerance of trans persons, as well as the stronger incidence of encountering homophobic words in the workplace, are likewise strongly associated with increased favor for trans protection guidelines amongst friends (Atteberry-Ash et al. 2018; Toomey et al. 2016). Furthermore, Phipps (2021) mentions in research on trans involvement in university athletics that the binary frameworks of sport might create accessibility challenges for all individuals, but mainly those who are trans. Undoubtedly, it is critical that sport be accessible to everyone, and the results suggest that a variety of initiatives need to be implemented to improve opportunities for athletics for the trans students.

Summary: The theme emphasizes how the heteronormative milieu of athletics may put additional pressure on transgender student athletes to perform and comply, increasing their likelihood of prejudice based on gender identity and decreasing their emotional health. Furthermore, sports' binary sex frameworks can pose accessibility issues for trans athletes. All these factors can act as a barrier for intercollegiate sports participation.

### 4.5. Strength and Limitation

This paper has several key strengths. Firstly, the scoping review technique enabled a thorough examination of the available literature regarding transgender social inclusion within college sports. This includes a variety of study types, populations investigated, as well as outcomes measured, allowing for a comprehensive overview of the current state of research on this issue. Additionally, this paper uses current sources (2013–2023) to analyze the issue; hence, it provides up-to-date information. Secondly, the scoping study revealed many research gaps in the literature, including the requirement for greater research on the perspectives of transgender athletes at the collegiate level and the usefulness of programs and policies targeted at increasing social inclusion. Furthermore, this paper identified some key barriers to the inclusion of transgender people such as lack of support, poor policy, stigma, and poor environment, where additional improvements are needed. Thirdly, the search method was extensive, utilizing a variety of keywords, sources, and search strategies to identify relevant papers.

Nevertheless, this paper has some key limitations, which need to be acknowledged. The search criteria are limited to 2013–2023; hence, it is possible that some of the important studies might be excluded. In addition, only papers that discuss transgender issues were selected. Secondly, the search approach was restricted to English-language publications, which may have omitted important articles published in other languages. Thirdly, since the scoping review was confined to peer-reviewed journals, it is possible that relevant studies conducted by non-peer-reviewed sites were overlooked. Fourthly, the research reviewed in the scoping review was diverse in terms of research design, demographic investigated, as well as outcomes assessed, potentially limiting the capacity to evaluate and integrate findings across research. Lastly, the scoping review did not analyze the validity and reliability of the reviewed research, which could have impacted the robustness of the review's results.

Altogether, the scoping review offered a thorough summary of the available research on transgender social inclusion within intercollegiate athletics. While the evaluation had several limitations, the merits exceeded the weaknesses and offered a good platform for future study in this area.

## 5. Conclusions

This scoping review gives a synopsis of the available research on transgender social inclusion in college sports. Despite the existence of a growing corpus of study on the subject, it is indicated that transgender people experience significant bias, which impacts their inclusion in sports events. Another significant issue is the lack of support. Transgender

people receive fewer amounts of organizational support compared to their peers. Moreover, most of the time transgender athletes feel insecure regarding their participation in an event. However, there are still substantial gaps in the current understanding of transgender athletes' perspectives at the collegiate level, in addition to the efficacy of programs and policies designed to promote social inclusion.

Future studies might look at the realities of transgender athletes at the collegiate level, such as their beliefs about the societal and cultural challenges they confront in games. This could better guide the creation of more comprehensive policies and initiatives targeted at fostering social inclusion as well as assisting transgender athletes to achieve their success.

Finally, additional study is needed on the long-term effects of social inclusion strategies and initiatives, as well as their influence on transgender athletes' mental and physical health, academic achievement, and future career chances.

Ultimately, this scoping review emphasizes the importance of the continuing study of transgender people's social integration in college athletics. We can foster a more equal and accessible sports culture for all participants by filling gaps in our knowledge on this critical topic.

**Author Contributions:** L.M.: Conceptualization, methodology, data curation, resources, writing—original draft, writing—review and editing; E.G.M.: Validation, writing—review and editing; L.L.: Methodology, validation, writing—original draft, review and editing; All authors have read and agreed to the published version of the manuscript.

**Funding:** This research received no external funding.

**Institutional Review Board Statement:** Not applicable.

**Informed Consent Statement:** Not applicable.

**Data Availability Statement:** All data generated or analysed during this study are included in this published article.

**Conflicts of Interest:** The authors declare no conflict of interest.

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
