# Peer review of "Social Inclusion of Transgender People in Intercollegiate Sports—A Scoping Review"

_socsci, doi:10.3390/socsci12060335_

Round 1

Reviewer 1 Report

Thank you for giving me the opportunity to review this paper. Unfortunately, I can’t recommend it for publication. I’ve made sure my feedback is as detailed as possible to explain why. Major changes are needed to the introduction with some smaller suggestions for the abstract and methods. However, the findings are unclear, confused, at times completely irrelevant to the topic, and are not clearly applied to intercollegiate sport, which is supposed to be the focus of the study.

My overall advice would be to re-visit the aim of the study and ensure this is much more focussed, possibly focussing only on trans women’s inclusion in sport (at present you switch between trans men, trans women, LGBTQ+ people, and issues around sexual orientation, even though these issues/barriers are wide ranging and diverse). The findings are therefore far too broad and unfocussed at present, with confusion/conflation of issues related to sexual orientation and gender identity (these are different). Once you’ve narrowed the focus, a clear contribution is needed linking this to the intercollegiate setting.

Abstract:

This is clear, but should generally be written in past tense, e.g., you have already located, assessed, and extracted the information from literature.

Introduction:

There are some terminology issues here. Sexual orientation and gender identity are different parts of someone’s identity. However, the first sentence confuses this. Sexual orientation refers to whether someone is gay, lesbian, bisexual or another sexual identity, and this is unrelated to your study (you don’t need to refer to this). Gender identity, on the other hand, refers to whether someone identifies as a man, woman, non-binary, or in another way. I would suggest avoiding the terms male/female when discussing gender as these usually relate to ascribed biological sex (different to gender identity). I think you need to be clear that as a society, we ascribe someone’s gender based on their biological sex, e.g., there is an assumption a child born female will identify as a girl/woman and with the societal constructions of femininity. Being trans is when there is an inconsistency between someone’s ascribed biological sex and the gender in which they identify.

You should use the terms ‘gender-affirming surgery’ and ‘gender-affirming hormones’. The term cross-sex hormones is considered outdated. In this same sentence, physically transition makes more sense than physical transition.

I think you also need to briefly outline what a social transition is, e.g., adopting clothing, hairstyles, change of name, mannerisms, etc., to better correspond with someone’s gender identity, without medically transitioning.

You give some context on discrimination towards trans people which is great. However, there is little information on the sport context. It may seem obvious but I would suggest highlighting the gendered nature of sport and the argument that in gendered spaces, we tend to see heightened levels of discrimination towards trans people (where does a trans person ‘fit’ and how does this impact their sense of belonging in these spaces?) We tend to structure sports based on binary models of sex/gender, e.g., the structure of two separate categories whereby men and women are rarely able to compete against one another (even in mixed sports there are still often gender-based rules). I think a paragraph is needed to provide this context before moving into the IOC policies.

The paragraph on the IOC policy needs developing. There were numerous issues with the 2004 policy (e.g., gender affirming surgery has no bearing on sport performance; the requirement of 2 years of gender affirming hormones was not based on evidence; there were the same requirements for trans men and women). You state there is a widely held idea that trans people have a competitive edge – I assume you are referring to trans women? Please make this clear. You state, ‘The National Collegiate Athletic Association (NCAA) is yet another body that has made it clear that transgender college athletes are welcome.’ This makes it seem like most sport bodies are welcoming of trans athletes. With the recent ban by World Athletics/RFU of any trans women at the elite level, I’d argue many are not welcoming at all. I’d suggest re-wording. Please also change to gender affirming hormones rather than hormone therapy here. In addition, the IOC no longer has a 2 year requirement – they now leave it up to individual national governing bodies to create their own rules specific to their sport. The update in policy needs to be explained here. Finally, you state ‘Nevertheless, there seems to be a significant difference between the rights and rules applicable to male along with female transgender athletes (Kondareva 2019).’ What exactly do you mean by this? At present, this isn’t clear.

Paragraph starting ‘although there are promising indicators’ on line 73– I would suggest deleting. It doesn’t flow clearly from the previous paragraph. You discuss disability and ethnicity which are unrelated to your study. You use the term LGBTQ but your study specifically focusses on trans people. The following paragraph (line 86) you then go back to discussing wider discrimination outside of the sport context (e.g., clothing stores, restrooms) which seems disjointed. I think a restructure of the intro is needed – start with wider discrimination, then the sport context, followed by the rationale for your study.

Maybe I missed this, but what is your research question/aims and objectives of the study? I’d advise making this clear at the end of the intro.

Methods:

Detailed and clear overall but a few minor points. You say a scoping review ‘would be’ applicable (line 120). Write in present or past tense – it IS applicable.

I would suggest avoiding saying any framework is ‘perfect’ – line 148.

Line 165 – this LED to 45 papers (past tense) – please go through the whole section to ensure consistency with the tense you use.  

Line 176/177 – three papers were published in 2019/2020/2021 (not one).

Line 187 – I would avoid saying ‘factual evidence’ – consider changing to primary/empirical data.

Can you make it clear if you were looking at social science based research? It seems you were, but then one article by Handelsmen et al., (2018) is not, based on the description in the table.

Discussion:

At present, this is really confused and unclear. You are discussing sexual orientation/LGB people which is not relevant to your study. You mention racism in your first subheading – but nothing you discuss clearly links to racism. The scoping review was supposed to be about intercollegiate athletics and yet the discussion seems to focus on everything related to LGBTQ+ people in sports (sexual orientation, discrimination, stereotyping, policies…). Only 4.4 seems to link to the collegiate setting and this is minimal drawing mainly from a study by Atteberry-Ash which doesn’t specifically focus on trans people. You are missing key literature relevant to the university/college setting (see suggestions below).

Literature you have missed which is very relevant to your study (in particular the first and last one)

https://journals.sagepub.com/doi/full/10.1177/1012690219889621

https://journals.sagepub.com/doi/10.1177/10126902231162270

https://www.tandfonline.com/doi/full/10.1016/j.smr.2018.09.006

Round 2

Reviewer 1 Report

I've also noted in addition to the below that the abstract is different on this page to the one I reviewed in the PDF document, so consistency is needed here. 

Overall, this is a significant improvement from the first submission. However, there are still a number of issues that need addressing. Please also do a thorough proofread of the work to minimise any grammatical errors or wrong word choice. For instance, in some of the changes you made, you have still used ‘hormone therapy’. With the additional changes, it also seems like there is a lack of flow between ideas/jumping between topics. I’ve provided some specific examples below (alongside other changes) but please go through the whole document for flow/continuity.

I know it is frustrating to do more major changes. This has the potential to be a really good study, but it isn’t there yet.

Abstract – I would change the last line back to present tense – ‘the results of the study are expected to inform’ but otherwise this looks good.

Intro – ‘Some of the difficulties they face include an absence of access to amenities that reflect their sexual orientation’ – you need to change sexual orientation to gender identity. I would also avoid the term ‘therapies’ which indicates a trans person has something wrong with them (you use this term again later several times).

‘A trans woman who has had hormone therapy, for instance, might have less testosterone level compared to cisgender women, yet greater levels compared to cisgender males, raising concerns regarding her ability to play female sports’ – I think you’ve got this sentence confused – do you mean a trans woman might have less testosterone than cisgender men, but still greater levels compared to cisgender women? Could you also critique this? Although this is often used as an argument against trans women, testosterone is not the only factor which impacts sport performance, and a lot of the literature that claims trans women have an advantage is flawed – see recent report here: Transgender Women Athletes and Elite Sport: A Scientific Review | Canadian Centre for Ethics in Sport (cces.ca)

In relation to the above, you need evidence for this statement: ‘Trans women could benefit from a competitive advantage within sports because of earlier exposure to testosterone hormones, which might confer some physical benefits like enhanced bone density, muscle mass, and increased oxygen-carrying ability.’ It also isn’t clear if you are talking about before or after a medical transition?

You also jump back and forth between different ideas a lot in this whole section – for instance you discuss the IOC policy, you then discuss how it has been amended, then you go back to the point that the 2004 policy had influence on other sport organisations – all valid points but very little flow between ideas. Later, you discuss the IOC policy then NCAA policy then back to IOC – the work is therefore very difficult to follow. You need to go through the whole document to ensure flow between ideas.

If you are going to mention the recent restrictions by World Athletics/RFU, this needs to be referenced and explained (even if you do this briefly or direct the reader to a recent report about this). At the moment, there is no context here and some readers might not know who the RFU are (rugby football union in England).

Discussion – this section should be based on the findings from the 14 articles in the table. References to other articles need removing or clarity is needed over why you’re adding additional sources in here. Why weren’t these sources included in the initial search if they are relevant?

I would also suggest you need to contextualise everything to the collegiate context. A lot of the literature you use isn’t relevant to collegiate sports, so having some sort of summary at the end of each theme stating that we can’t always draw strong conclusions to the collegiate context is needed – however, based on literature in other areas (e.g., leisure sports, non-collegiate competitive sports) we CAN begin to understand trans inclusion sport issues that MIGHT also be applicable to the collegiate context. What is unique about this context which might not be addressed in other sports literature?

In the theme ‘policy issues’ you state ‘The issue is that the NCAA's one-year suppressor rule is not rigorous enough to establish an equal playing field among Thomas as well as the cisgender females she is competing against’ citing what looks like a non-peer reviewed source (Saeger, 2022). You don’t have the evidence to state this with certainty. Based on the fact Thomas didn’t break any records (NCAA, state or US-based), and only won one race (coming 8th and joint 5th in the other two races at the same meet) – see here Lia Thomas finishes 8th in 100-yard freestyle, final race of collegiate swimming career (espn.co.uk), do we really know that Thomas has an advantage? Her race times indicate she sits within times we’d expect for elite women.

Section 4.3 is not developed enough and seems to draw on a lot of literature not in the table.

Section 4.4 is titled ‘issues in collegiate settings’. Your whole study should be based on the collegiate setting, so why is there a distinct theme on this? Issues in the collegiate setting should be evident in each findings section and shouldn’t be a unique theme.

‘Additional research examining heterosexual student-athletes trans friend’ – what do you mean by this?

Reviewer 2 Report

Thank you for the opportunity to review this manuscript. Overall, I found the paper to be improved from the first iteration and I especially appreciated the labeling of the study types for each article in Table 1. However, I do not yet think the manuscript is ready for publication as there remain areas of improvement or issues that need to be addressed. In particular, I think there are two areas: (1) the focus of the paper needs to continue to be narrowed to focus specifically on US-based college athletes, and (2) the overall flow of the manuscript is inconsistent or abrupt at times, to the point where the main argument(s) and/or observation(s) are lost or mitigated. To more effectively provide points of clarification or revision, I have organized my feedback by paper section:

Abstract

·      Rather than dedicating several words to the inclusion criteria for articles (this is not the most relevant takeaway/element from the study, particularly for a short abstract, in my opinion), I would recommend using this space to signpost your findings, ideally before stating the broader impacts of the findings.

Introduction

·      The paragraph beginning on p. 1, line 43 (“The subject of……(Westbrook and Schilt 2014)): the flow of this paragraph is not coherent. Namely, the topic sentence indicates that intercollegiate sport will be discussed, yet, the paragraph includes a discussion of prohibiting sports competition in different nations, (depending on national law). While I do not disagree with the facts and observations in this paragraph, it is not specifically linked to intercollegiate sport. A possible way to revise this flow issue is to start broadly (i.e., the overarching challenges that trans folks face in sporting/physical activity spaces), then narrow to college athletics. As well, I assume this is specific to US-based college athletics?

·      In a similar vein, I would recommend moving the paragraph that starts on p. 2, line 94 (“Sport constitutes a highly…against trans athletes”) to earlier in the introduction. The overarching discussion of sport as a gendered environment is fundamental to trans “debates,” as the author/s point out, but should be made earlier in the section for reading flow.

·      P. 2, line 107-108 (“Trans athletes could feel alone…”): what is the reference for this claim?

·      P. 2, line 114 (“…strict and old-fashioned gender stereotypes”): be careful making these kinds of characterizations without references, particularly in a review piece.

·      The discussion of the IOC’s policy starting on line 116, p. 2 still reads as slightly out of place and the jumps between IOC and NCAA are a bit jarring. I am not convinced that this background necessary and think that this space can be better used to focus on the development of NCAA policy (past, but especially present) to help narrow the focus of the manuscript.

·      The transition on p. 3 from lines 167 to 168 is abrupt (“Gyms are indeed popular…exercise and sports. Firstly, transgender…”), please make this transition smoother.

Method

·      P. 3, line 190: should “Social” be capitalized?

·      Section 2.6: I believe it should be “Microsoft Excel was used” to match the verb tenses within the rest of sub-section.

·      P. 6, line 323: “…for the current stages”: the current stages of what? Please be specific.

Discussion

·      Overall, the findings should be located within the collegiate sport setting, particularly since the author/s note that their contribution(s) to the existing literature is specific to the college sport landscape. This was particularly notable for section 4.3 (which read as a bit under-developed and not enough inclusion of the articles listed in Table 1) and section 4.4 (which is specific to college sport…which seems at odds with the overarching purpose of the manuscript, which the author/s state is to focus on collegiate sport).

·      P. 8, lines 503-505: I do not disagree with the sentiments in here, but there should be a bit more explanation. How/In what ways do “considerable impediments” (for trans athletes) remain in sport, as a result of the gender binary? How/Where do eligibility concerns and transphobia fit within obstacles raised by the gender binary?

·      P. 8, line 521: “racism against trans people is significantly high” I don’t doubt this, but there should be more contextualization. How/Why is this the case?

·      P. 9, the discussion of Lia Thomas, particularly “the NCAA’s one-year suppressor rule is not rigorous enough to establish an equal playing field among…”: I would heavily caution the author/s from making this claim given the broader context of Thomas’ times, places, and records. In a similar vein, there is a robust body of literature dispelling the myth of “an equal playing field,” in any competitive category.

·      Page 10, line 717: “heterosexual student-athletes trans friend sentiments”: unclear what is meant by “friend sentiments,” please clarify.

In closing, I want to reiterate that I believe the topic of this paper to be timely and important, and of interest to Social Sciences readers. I commend the author/s in their first round of revisions and hope that my comments are helpful in moving the manuscript forward.
